# Assessment of Carbon Storage under Different SSP-RCP Scenarios in Terrestrial Ecosystems of Jilin Province, China

**DOI:** 10.3390/ijerph20043691

**Published:** 2023-02-19

**Authors:** Daiji Wan, Jiping Liu, Dandan Zhao

**Affiliations:** College of Tourism and Geographical Sciences, Jilin Normal University, Siping 136000, China

**Keywords:** SSP-RCP scenarios, carbon storage, land use change, PLUS model, InVEST model

## Abstract

Carbon storage is one of the key factors determining the global carbon balance in the terrestrial ecosystems. Predicting future changes in carbon storage is significant for regional sustainable development in the background of the “dual carbon” objective. This study which coupled the InVEST model and the PLUS model and is based on land use in different future scenarios evaluated the evolution characterization of terrestrial carbon storage in Jilin Province from 2000 to 2040 and explored the impact of related factors on it. The results show that: (1) from 2000 to 2020, the area of cultivated land and built-up areas increased continuously in Jilin Province, while the area of forest land, grassland, and wetland decreased with time; the ecological land has been restored to a certain degree. (2) Due to the continuous reduction in ecological land, the overall carbon storage in Jilin Province from 2000 to 2020 showed a downward trend, with a total reduction of 30.3 Tg, and the carbon storage in the western part of Jilin Province changed significantly. The SSP2-RCP4.5 scenario shows a minimum value of carbon storage in 2030 and a small increase in 2040; the SSP1-RCP2.6 scenario shows an increasing trend in carbon storage from 2020 to 2040; the area of built-up areas and cultivated land increases and the loss in carbon storage is more serious under the SSP5-RCP8.5 scenario. (3) On the whole, with the increase in elevation and slope, the carbon storage showed a trend of increasing first and then decreasing, and the carbon storage of shady and semi-shady slopes was higher than that of sunny and semi-sunny slopes; forest land and cultivated land were the keys to carbon storage changes in Jilin Province.

## 1. Introduction

Reducing carbon emissions and improving the storage capacity of carbon have become critical issues for scholars in various countries with the proposed “carbon peaking and carbon neutrality” goals [1]. The terrestrial ecosystem serves as a significant component of carbon storage in the world. Among it, land use change is not only the main manifestation of human activities but also a key factor driving regional carbon storage changes on different scales [2]. The terrestrial ecosystem influences the climate system through the exchange of carbon between their subsystems and the atmosphere. At the same time, carbon storage increases in the terrestrial ecosystem by capturing and storing carbon in the atmosphere and reducing atmospheric carbon dioxide concentrations [3]. Additionally, the loss in terrestrial carbon storage leads to an increase in atmospheric CO_2_ concentration, which leads to an acceleration in the global warming process [4]. Therefore, it is important to investigate and predict the effect of regional land use change on terrestrial carbon storage to improve the regional ecosystem service capacity and reduce carbon emissions [5].

The study of CS (carbon storage) changes affected by LUCC (land use/cover change) plays a pivotal role in maintaining the stability and balance of the carbon cycle in the terrestrial ecosystem [6]. Many studies have been conducted to estimate the carbon storage of different land categories via remote sensing model inversion [7,8,9] and sampling surveys in the field [10,11,12]. However, the LUCC-based estimation of carbon storage and prediction is gradually being used widely [13]. In recent years, many scholars have carried out research work on carbon storage estimation on different regional scales, such as cities [2,14,15], drainage basins [13,16], provinces [17,18], and countries [19,20]. Some scholars have also estimated and projected carbon storage under different scenarios [5,21] and different policies [1,5,22]. China is a country with a vast territory. It lacks the efficiency to obtain field measurement data over large areas via field sampling and other means. In contrast, the InVEST (integrated valuation of ecosystem services and trade-offs) model has been widely used for ecosystem service value studies, such as biodiversity and present and future carbon storage, and has shown excellent applicability in studies in domestic and international regions [6]. Currently, the majority of studies focus on carbon storage changes in a single land type. Most of the studies have been conducted to predict and analyze temporal changes. The spatial heterogeneity of carbon storage and how the drivers affect carbon storage still need to be further studied. The quantification of the relation between carbon storage and land use, thus enabling the study of spatial divergence of carbon storage, has also become a vital issue in carbon storage research. Zhu et al. analyzed the influence of LUCC on carbon storage in the Chinese arid zone using CA–Markov models, and the results showed that grassland degradation and agricultural land expansion were the main causes of carbon storage reduction in the arid zone [16]. Zhang et al. estimated carbon storage in Guilin, China, by using the FLUS model, and the results showed that spatial-temporal differentiation is significant in carbon storage levels within the region. The regional carbon sequestration capacity increased after the adoption of resource conservation measures [4]. Zhu et al. assessed the CS (carbon storage) in the Qihe catchment by using the CLUE-S model and the results showed that the expansion in built-up areas and the degradation of forest land under historical and future scenarios are the leading causes for the decrease in carbon density [5]. The PLUS model (patch-generating land use simulation model) which was developed by the HPSCIL@CUG laboratory development team of the China University of Geosciences applies a new strategy of land expansion analysis to explore the causal factors of various land use changes better and it is used extensively for large-scale simulations of land use.

The IPCC (Intergovernmental Panel on Climate Change) [23] proposes new prognostic scenarios, SSP-RCP, based on the recent anthropogenic emission trends and different shared socio-economic pathways (SSPs). The scenarios aim to describe potential development pathways under diverse emission and socio-economic contexts and to assess the societal capacity to address mitigation and adaptation challenges [24]. Different SSP-RCP scenarios can be used to assess global emission reduction goals and different policies for future LUCC and highlight the role of different socio-economic development patterns in driving climate change. It is better to predict the CS in the future by combining SSP-RCP scenarios into the estimation and simulation of terrestrial ecosystem carbon storage. It is significant to explore the relationship between CS and climate change, and to achieve the “dual carbon” objective and sustainable development.

Jilin Province has an indispensable status in China’s geographical and ecological pattern and modernization development process. The eastern region of Jilin Province is the Changbai Mountains, which is an essential ecological barrier in northeast China. The west of Jilin Province is an important commercial grain production area and there is a concentration of saline marsh wetlands in China. The topography of Jilin Province is relatively undulating, and the morphology of the landscape varies significantly. The main land use categories in Jilin Province are cultivated land and forest land, showing a landscape of “western cultivation and eastern forestry” [25,26]. With the interaction of climate change, population growth, and accelerated urbanization, the terrestrial ecosystem in Jilin Province is changing more drastically. This work couples the PLUS model with the InVEST model to investigate the spatial-temporal partitioning and evolutionary characteristics of carbon storage in the future terrestrial ecosystem of Jilin Province under different SSP-RCP scenarios. This study analyzes the effects of LUCC and topographic factors on carbon storage. The purpose is to provide comments and references for land resource management, carbon emission reduction, carbon storage enhancement, and sustainable development in the region.

## 2. Materials and Methods

### 2.1. Study Area

Jilin Province (122°38′ E–131°19′ E, 41°52′ N–46°18′ N) is located in the central part of northeast China (Figure 1). It is bordered by Russia and North Korea to the southeast, and by Heilongjiang Province, Inner Mongolia Autonomous Region, and Liaoning Province to the north, west, and south, respectively. The total area is 18.74 × 10^4^ km^2^ [26]. Jilin Province has eight prefecture-level cities, including Changchun, Jilin, Siping, Tonghua, Baishan, Liaoyuan, Baicheng, and Songyuan, and the Yanbian Korean Autonomous Prefecture [25]. There are obvious seasonal variations and geographical differences in temperature and precipitation in Jilin Province, and the monsoon climate is significant [25]. The topography of the region slopes from the mountains in the southeast to the plains in the central and western parts, and shows an obvious spatial characteristic of “high in the southeast and low in the northwest” [26]. The study area spans five major water systems: Tumen River, Yalu River, Liao River, Suifen River, and Songhua River, and the rivers are distributed in the region unevenly. Land use changes in Jilin Province are more significant due to the impact of climate change and human activities.

### 2.2. Data Source and Processing

The data used for a future simulation of land use in this study involved historical land use data and meteorological data, socio-economic data, elevation data, basic geographic information data, and other driver data (Table 1). The land use data were acquired from Landsat images via random forest algorithm classification and manual visual interpretation, and 15% of each land category was selected for random sampling. The kappa coefficient and overall accuracy were calculated as 0.887 and 0.912 which indicates a high consistency of distribution among land categories to support research needs. Historical land use classification was reclassified into 6 categories, which included cultivated land (dryland and paddy field), forest land (forested land, shrubland, sparse forest land, etc.), grassland (high-, medium-, and low-cover grassland, etc.), wetland (reservoir pits, lakes, rivers, beaches, etc.), built-up areas (urban and rural settlements, transportation land, industrial land, etc.), and other lands (sandy land, swampy land, saline land, bare land, etc.). The future land use was simulated using land use data and future land demand extracted from the Land-Use Harmonization (LUH2) dataset under different scenarios. Socio-economic data, such as population density, were obtained via interpolation of panel data. These drivers were made into raster data with the same number of rows and projection coordinates to be used in the simulation of the PLUS model in this work (Figure 2). Future climate data were obtained from the NEX-GDDP-CMIP6 dataset at downscale, and socio-economic and demographic data were obtained from the RCP dataset. Slope and aspect were extracted from the elevation data. The spatial resolution of the raster data used in this work was inconsistent, which was uniformly resampled to 100 × 100 m with full consideration of the extent of the study area and the operation of the prediction model. The carbon density was gained from existing studies by other scholars in this work [27,28] and the selected data were verified against the 2010 Chinese Terrestrial Ecosystem Carbon Density Dataset created by Xu et al. [28] and other scholars’ studies in areas which overlap or are geographically similar to Jilin Province. The results were in a reasonable range and were also corrected using the correcting equation.

### 2.3. Method of Study

#### 2.3.1. PLUS Model

The PLUS model is a CA-principle-based land use change simulation model that couples a new land expansion analysis strategy (LEAS) with a CA model based on multiple types of random patch seeds (CARS) to predict and simulate the generation and evolution of multiple land category patches [29]. The model extracts expansive lands through multi-period historical land use data. The expansive lands are sampled regularly or randomly by an RF (random forest) algorithm with driving factors. The development probability of each land use category is calculated one by one [30]. The equation is as follows.
(1)$$P_{i,k\left(x\right)}^{d}=\frac{\sum_{n=1}^{M}I\left(h_{n}\left(x\right)=d\right)}{M}$$

In Equation (1), *d* = 0 indicates that other land categories do not shift to land category *K*, while *d* = 1 can shift to land class *K*. *x* is a vector consisting of multiple drivers; *I*(·) is the indicator function of the set of decision trees; *h_n_*(*x*) is the predicted land category with a decision tree of *n*; *M* is the total number of decision trees. Domain weights (*Ω*) are used to show the strength of the ability of various land use categories to expand or transform into other land use categories, with the following formula:(2)$$\Omega_{i,k}^{t}=\frac{con\;\left(c_{i}^{t-1}=k\right)}{n\times n-1}\times w_{k}$$

In Equation (2), *W_k_* denotes the domain weight coefficient of land category *k*, which takes the value of [0, 1]. It characterizes the strength of the ability of a land category to expand to other land categories. $con\;\left(c_{i}^{t-1}=k\right)$ is the total number of cells occupied by the *n* × *n* meta cells iterated to the final time for the land category *k* [31]. In this work, one refers to the neighborhood parameter settings of other related studies and sets five sets of neighborhood parameters with the actual condition of Jilin. Different neighborhood weight parameters were inputted to simulate the land use of the study area in 2020. The output results were compared with the real condition in 2020 for verification, and the best set of output results was selected as the neighborhood weight parameters for this study (Table 2). The random patch generation mechanism of the PLUS model can simulate the natural growth of land categories during land use change better [32]. In this work, nature reserves and other areas were set as restricted areas for land use conversion in consideration of the policy in China.

The confusion matrix and Fom module of the PLUS model was used to assess the consistency between simulation results and actual land use distribution. This study simulated LUCC and spatial pattern in 2020 by using the land use data of Jilin Province in 2000 and 2010 as the beginning and end years. The accuracy was verified with the actual results. The validation results show that the kappa coefficient and overall accuracy reached 0.891 and 0.932, respectively. Both of them are greater than 0.75, which indicates that the credibility of the results is high. Considering factors such as geographic environment, current socio-economic development, and developed policies in Jilin Province, three development scenarios were set up to predict the LUCC of the future in this study as follows.

(1)SSP1-RCP2.6 scenario: This scenario can be understood as a sustainable development scenario, which strictly limits the transfer out of ecological land such as forest land and grassland by setting constraints, i.e., strengthening the protection of ecological lands. At the same time, relevant policies and plans are considered, and a cost transfer matrix with ecological protection as the priority is set without affecting the urbanization process, i.e., without reducing the expansion capacity of built-up areas and other lands. This scenario shows a trend of population increase and then decrease from the beginning of the 21st century to 2100, and a clear trend of GDP increase. Forest land is protected under this scenario and the type of economic development is dominated by forestry.(2)SSP2-RCP4.5 scenario: This scenario represents an intermediate path with moderate radiative forcing, which is a continuation of the current land use change trend. The land demand in the LUH2 dataset is extracted to obtain the number and area of each category of raster in Jilin Province in 2030 and 2040 under this scenario. The future land use changes in Jilin Province are simulated without setting any constraints under this scenario. The trend of population and GDP change in this scenario is the same as in SSP1, but the difference is that the turning point of population change occurs in 2030. The type of economic activity shows a trend in the synergistic development of various industries such as agriculture, pastoralism, and forestry.(3)SSP5-RCP8.5 scenario: Different from the above scenario, this scenario prioritizes economic development. High economic development is achieved through the continuous development of fossil fuels. Meanwhile, SSP585 is the scenario with the highest greenhouse gas (GHG) emissions. The high-emission development strategy leads to substantial changes in various land categories. The turning point of population change in this scenario occurs in 2025, and the trend of change is basically the same as in SSP1. The type of economic activity is dominated by industry.

#### 2.3.2. InVEST Model

The carbon storage and sequestration of the InVEST model (integrated valuation of ecosystem services and trade-offs) is an effective tool for estimating terrestrial carbon storage [18]. It divides terrestrial carbon storage into four principal carbon pools, which are AGC (above-ground carbon storage), BGC (below-ground carbon storage), SOC (soil carbon storage), and dead organism carbon storage (Equation (3)) [33].
(3)$$C_{i}=C_{i-above}+C_{i-below}+C_{i-soil}+C_{i-dead}$$
(4)$$C_{total}=\sum_{i=1}^{n}C_{i}S_{i}$$

In Equations (3) and (4), *C_total_* is the total terrestrial carbon storage, *C_i_* represents the total carbon storage of land category *i*, *C_above_* refers to above-ground biotic carbon storage, and *C_below_* represents below-ground biological carbon storage. *C_soil_* means soil carbon storage with soil depth of 10 cm. *C_dead_* indicates the carbon storage of dead organisms. *S_i_* is the total area of land category *i* [17]. 

Due to the low carbon density content of dead organisms and the difficulty of obtaining data, only the carbon storage of three major carbon pools was studied in this work. To ensure the accuracy of the study results, carbon density data within Jilin Province were used as much as possible, and carbon density values obtained by the same scholar or the same method were used due to the difference in research methods that may have caused deviations in the results [34]. The data collected through previous studies were all carbon density data from the 2010s. However, carbon density has the characteristics of changing with time and climate change [13]. To enhance the accuracy of carbon storage simulation results, it is necessary to average the carbon density values of secondary land categories with little difference and correct the carbon density by using meteorological factors.

The results of domestic and international studies show that both biological carbon density and soil carbon density are positively correlated with annual average precipitation [35,36], and most of the established studies have obtained correction coefficients by comparing other regions with clear historical carbon density with the study area [2]. Combining the established studies with the needs of this work, the following equations were selected for the correction of carbon density.
(5)$$C_{sp}=3.3968\times P+3996.1\;\left(R^{2}=0.11\right)$$
(6)$$C_{bp}=6.7981e^{0.00541P}\;\left(R^{2}=0.7\right)$$
(7)$$K_{sp}=C_{sp1}/C_{sp2}$$
(8)$$K_{bp}=C_{bp1}/C_{bp2}$$
where *C_sp_* is the annual precipitation-corrected soil carbon density data (unit: Mg/hm^−2^); *C_bp_* is the annual precipitation-corrected biological carbon density data (unit: Mg/hm^−2^); and *P* is the annual average precipitation (unit: mm). *K_sp_* is the annual average precipitation correction factor of soil carbon density. *K_bp_* is the annual average precipitation correction factor of biological carbon density; *C_sp1_* and *C_bp1_* are the carbon density correction factors of Jilin Province. *C_sp1_* and *C_bp1_* are the correction coefficients for carbon density in Jilin Province. *C_sp2_* and *C_bp2_* are the correction coefficients for historical carbon density in Jilin Province. The values of the average annual precipitation are 766.6 mm and 811.2 mm, which are the average annual precipitation of Jilin Province in 2020 and of Jilin Province in 2010, respectively. The correction coefficients were substituted into the correction equation and revised to obtain the missing carbon density values for each land category. Considering the difficulty of obtaining carbon density data and the deviation in predicted values, the constant carbon density values selected were used to calculate the changes in carbon storage in historical and future periods in this work (Table 3). The partial carbon density data of 2020 obtained via correction were compared with the carbon density data collected in the field for verification. The measured above-ground carbon density of grassland in western Jilin Province is 19.8 Mg·hm^−2^, and the soil carbon density is 330.2 Mg·hm^−2^; the soil carbon density of wetland is 141.7 Mg·hm^−2^, and the soil carbon density of other lands is 248.1 Mg·hm^−2^. In a reasonable range, the results can be used to input model parameters.

## 3. Results and Analysis

### 3.1. Land Use Dynamics and Multi-Scenario Predictions

The major land categories in Jilin Province from 2000 to 2020 are forest land and cultivated land. They cover 84% of the total study area, and both show an increasing trend. Wetland accounts for about 2.1–2.5% of Jilin, and shows an overall trend of decrease, with a reduction of 0.3%; built-up areas increase by 1463.92 km^2^, with an increase of 0.8%; grassland has shrunk sharply in the last 20 years, decreasing by 1228.04 km^2^, with a reduction of 0.7%; the area of other lands shows an increase and then a reduction in change, with a total decrease of 145.6 km^2^ (Table 4).

From 2000 to 2020, the cultivated land changed significantly in the eastern and southeastern parts of the province, and forest land was mainly converted into cultivated land. Due to the accelerated urbanization and human activities, the forest land in the eastern and southeastern parts of Jilin Province was reduced, while the forest land was relatively increased in the western and central parts of the province. The growth of the built-up areas in Jilin Province appears in the urban areas mainly where municipalities and counties are located, and most of them show a dispersion of growth in every direction, and the area of built-up areas in Jilin Province has substantially increased over the last 20 years. Other lands are concentrated in the western part of Jilin Province, and a small amount of other lands also exists in the eastern part, which is mainly converted into grassland and cultivated land in the western part of Jilin Province; other lands are mainly converted into forest land and cultivated land in the eastern part. Overall, LUCC in Jilin Province is characterized by the growth in built-up areas and cultivated land, with a continuous decrease in grassland and other lands.

The characteristics and trends in spatial-temporal land use changes in Jilin Province from 2020 to 2040 have significant variability under different scenarios. Under the SSP2-RCP4.5 scenario, the turn-out area of other lands in Jilin Province is the largest, 1375 km^2^, accounting for 36% of the total transferred area; the minimum transfer-out area of built-up areas, only 43 km^2^, accounts for about 1% of the turn-out area; and the turn-out areas of grassland, forest land, and wetland are 1271.08 km^2^, 673.24 km^2^, and 438.6 km^2^, accounting for 33%, 18%, and 12% of the total transferred area (Table 5). Meanwhile, the transferred area of grassland and other lands under this scenario is higher, 1033.56 km^2^ and 1019.44 km^2^, respectively, which is about 54 % of the total transfer area; the transferred area of cultivated land and built-up areas accounts for about 24% and 21%, with 894.68 km^2^ and 815.64 km^2^, while the transferred area of wetland is the least at only 37.6 km^2^. In contrast, the conversion rate of ecological lands such as wetland and forest land is reduced significantly under the SSP1-RCP2.6 scenario. The converted area of grassland is 442.76 km^2^, accounting for 18% of the total converted area; compared to the SSP2-RCP4.5 scenario, the area converted from cultivated land increased significantly, accounting for about 66% of the total area converted. The converted area of built-up areas and other lands was 19.08 km^2^ and 355.56 km^2^, respectively, which together accounted for 16% of the total converted area. Meanwhile, the transferred areas of forest land, cultivated land, and wetland were 1570.36 km^2^, 347.32 km^2^, and 242.32 km^2^ under the SSP1-RCP2.6 scenario. Among these, the net transferred area of forest land is the largest, followed by wetland; the areas transferred into the area of cultivated land and grassland were smaller than the transferred-out area, and the overall trend was degradation. In the SSP1-RCP2.6 scenario, cultivated land still showed a shrinking trend, while forest land and wetland increased significantly (Figure 3). In contrast, the area of built-up areas and cultivated land increased under the SSP5-RCP8.5 scenario, while the forest land, grassland, and wetland relatively decreased. Compared to 2020, cultivated land will have increased by 1.16% in 2030 and 1.17% in 2040; the area of forest land will have decreased by 0.84% and 0.80% in 2030 and 2040, respectively; and the area of grassland will have decreased significantly under this scenario, decreasing by 6.87% in 2030 and 8.56% in 2040 compared to 2020. The wetland will decrease by 0.66% from 2020 to 2030 and by 9.46% from 2020 to 2040, showing an overall decrease in the wetland, and the built-up areas will increase by 9.51% and 13.01% in the two periods, which shows that the demand for production land and living land will increase in this scenario, while the ecological land will decrease, relatively. Comparing the area of other lands, it is found that the land use efficiency will increase significantly in 2030 and 2040 compared to the historical period. In summary, ecological land such as forestland and grassland increases significantly under the SSP126 scenario, while the land use changes under the SSP245 scenario are closer to the current development of Jilin Province, and the SSP585 scenario focuses more on economic development and neglects the protection of ecological land.

### 3.2. Carbon Storage Change Characteristics in 2000–2040

The carbon storage of terrestrial ecosystems in Jilin Province was 7.44 Pg, 7.45 Pg, and 7.41 Pg in 2000, 2010, and 2020, respectively (Table 6). The carbon storage increased by 4.5 Tg, and the CS in forest land and cultivated land were 3.88 Pg and 2.46 Pg from 2000 to 2010. The proportion of total carbon storage was 52% and 33%. The proportions of other lands, grassland, wetland, and built-up areas are 6%, 5%, 2%, and 2%. From 2010 to 2020, the carbon storage decreased by 34.0 Tg, because of the decrease in forest land and grassland areas mainly, and the proportion of CS in cultivated land increased while forest land and grassland decreased. The total carbon storage decreased by 30.3 Tg from 2000 to 2020, and the percentage of carbon storage in each category did not change, relatively. The SSP2-RCP4.5 scenario shows a 39.0 Tg loss in carbon storage in 2020–2030. The CS of forest land, grassland, wetland, and other lands shows different degrees of a decreasing trend. Additionally, the relative increase in carbon storage in built-up areas is due to the continuous expansion of cities. In 2020–2040, the terrestrial ecosystem will lose a total of 36.0 Tg of carbon. Except for cultivated land and built-up areas, all other land categories present a decreasing trend in carbon storage. The carbon storage will increase by 16.5 Tg in 2020–2030 under the SSP1-RCP2.6 scenario. There will be an increase in carbon storage in forest land, water, and built-up areas, and a decrease in carbon storage in cultivated land, grassland, and other lands. The total terrestrial carbon storage will decrease by 13.0 Tg from 2020 to 2030 under the SSP5-RCP8.5 scenario in Jilin Province. Biological carbon storage will decrease by 4.0 Tg and soil carbon storage will decrease by 9.0 Tg. The total carbon storage in 2040 will remain reduced and less than that in 2030 under this scenario. It can be observed that in the SSP126 scenario, the reasonable conservation of ecological lands such as forest land and grassland is the key to carbon storage growth. In the SSP245 scenario, the loss in carbon storage is caused by the rapid expansion of cultivated land and built-up areas, while the increase in human activity intensity and emissions in the SSP585 scenario is one of the causes of the decrease in carbon storage.

In terms of the spatial pattern of CS, as a whole, terrestrial carbon storage in Jilin Province is relatively high and presents obvious spatial differentiation characteristics. The spatial distribution of carbon storage in Jilin Province changed significantly from 2000 to 2020. The areas with relatively dramatic changes in carbon storage were mainly Changchun City, Baicheng City, Songyuan City, Baishan City, and Yanbian Korean Autonomous Prefecture. The main manifestations were the massive increase in the area of transferred cultivated land in the western and central parts of the province, the increase in forest land in the east, the increase in other lands in the west, and the expansion in built-up areas around the province (Figure 4). The SSP1-RCP2.6 scenario, from 2020 to 2030, shows an overall trend in increasing carbon storage in the eastern, southern, and central regions of Jilin Province, while the carbon storage in the western region still shows a decreasing trend. From 2020 to 2040, the carbon storage in the western part of Jilin Province will change significantly, mainly due to the transfer in and out of grassland and other lands in the western region, resulting in fluctuating changes in carbon storage. The loss in carbon storage in the western Jilin Province in 2020–2030 is more serious under the SSP2-RCP4.5 scenario, while the carbon storage in the central and western regions will change more in 2020–2040. The carbon storage in the central region shows a significant decreasing trend in 2020–2030 under the SSP5-RCP8.5 scenario; meanwhile, the carbon storage in the northwestern Jilin Province will increase. The carbon storage in the whole region shows an obvious decreasing trend for 2020–2040, and the increase in carbon storage will still occur in the western region. Overall, the carbon storage in the SSP1-RCP2.6 scenario changes less than that under the SSP2-RCP4.5 scenario. The carbon storage of the SSP1-RCP2.6 scenario is higher than that of the SSP2-RCP4.5 scenario in the same year due to the reasonable protection of ecological land with higher carbon density. Carbon storage as a whole shows SSP126 > SSP585 > SSP245 in 2030, while the change in carbon storage in 2040 shows SSP126 > SSP245 > SSP585.

### 3.3. Impact Factors of Carbon Storage Changes

There is a significant spatial differentiation between land use categories and topography in Jilin Province. The higher elevation areas in the east are mostly forest land, and human activities are less compared with those in the west of Jilin Province. Therefore, this work explores the impact of topographic factors and LUCC on carbon storage changes. In this work, three factors which are elevation, slope, and slope direction were selected to discuss the influence of topographic factors on the change in CS. According to the actual situation of Jilin, the elevation and slope of the study area were re-divided into six gradients, and the slope direction was divided into four gradients. On the premise that the land area of each gradient was similar, the elevation was reclassified as less than 150 m, 150–280 m, 280–480 m, 480–850 m, 850–1500 m, and more than 1500 m, and the slope was reclassified as 0–3°, 3–4°, 4–7°, 7–14°, 14–30°, and more than 30°. The slope direction was reclassified as shady slope (0–45°, 315–360°), semi-shady slope (45–90°, 270–315°), semi-positive slope (90–135°, 225–270°) and positive slope (135–225°). 

Statistical analysis of carbon storage under different gradients of the three factors (Figure 5) shows that the overall carbon storage in Jilin Province shows a tendency to first increase and then decrease with rising altitude. The carbon storage below 280 m above sea level shows a trend of increasing with elevation, while the carbon storage from 480–850 m rises sharply and peaks, and the carbon storage from 850 m-1500 m shows a precipitous decline, with a “minimum” value above 1500 m. The reasons for this trend in carbon storage at different altitudes are as follows: forest land, cultivated land, and built-up areas are mostly below 280 m above sea level; carbon storage relatively decreases because of the increase in human activities from 280 m to 480 m; there is high vegetation cover where there is mainly forest land because of fewer human activities from 480 m to 850 m; and from 850 m to 1500 m, carbon storage is relatively low because of the decrease in plant diversity and biomass in high altitude areas. The land area above 1500 m only accounts for 0.37% of the total area; so, the estimated carbon storage is low.

The CS fluctuates greatly with the increase in slope, the carbon storage volume decreases and then increases from 0° to 7°, and the “maximum” value appears from 7° to 14°. The reasons for this change in carbon storage in different slopes are as follows: the carbon storage total is low because the terrain below 7° is relatively flat and suitable for the development and utilization of built-up areas and cultivated land; the area above 7° has fewer human activities and high vegetation cover, so the carbon storage is higher; and the proportion of the area with a slope of more than 30° of the total area is too small, which leads to the low carbon storage measurement result. The carbon storage of different slope orientations differed to a certain extent, and the overall trend was shady slope > positive slope > semi-shady slope > semi-positive slope. The trends in biological carbon storage and soil carbon storage under the influence of the three topographic factors are basically the same as the total carbon storage and will not be repeated in this work.

Land use change has a relatively strong correlation with total biomass, carbon sequestration capacity, and vegetation cover of the terrestrial ecosystems. The huge differences in carbon density among different land categories lead to differences in the impact of land category shifts on carbon storage. This work quantifies the carbon storage under different scenarios from 2000 to 2040 (Figure 6) and finds that carbon storage changes are significantly correlated with land use categories. When the land categories with lower carbon density shift to those with higher carbon density, it is more favorable for carbon sink formation and carbon storage increases. The land category in the eastern and southeastern regions of Jilin Province is dominated by forest land. The west and central regions are dominated by cultivated land. Because of the difference in carbon density between the two land categories, the overall carbon storage shows a spatial distribution of “high in the southeast and low in the northwest”. The carbon storage of forest land always accounts for more than 50% of the total CS, and the carbon storage of cultivated land accounts for more than 30%, so the changes in forest land and cultivated land largely determine the changes in carbon storage in Jilin Province.

## 4. Discussion and Conclusions

### 4.1. Discussion

The estimation of carbon storage based on land use is in this work, because the process of LUCC has a certain complexity and is influenced by a variety of factors such as natural changes and human activities. Although many factors were selected in this work to simulate land use change, the consideration of the influence of relevant policy factors is slightly lacking. Different development scenarios are provided in this work for future simulation, and the driving effect of climate scenarios on future land use change is not fully considered. Due to the limitation of carbon density acquisition, all of the land categories in this work used the first-class land category, but the variability in carbon density among the same land use types has some influence on the final estimation results.

Constrained by natural factors and geographical conditions, the urbanization process and economic development level in each region in Jilin Province show significant differences. From 2000 to 2020, the drastic expansion of built-up areas in all regions of the province led to a reduction in carbon storage. The central region of Jilin Province has a relatively rapid urbanization process which has resulted in the greatest change in CS. The eastern and southeastern parts of Jilin Province are relatively high in elevation and the terrain is mainly mountainous; so, the economic development in this region is limited and the transfer of built-up areas is relatively inefficient. In addition, the transfer of ecological land is one of the main reasons for the decline in CS, as forest land is encroached by land categories with relatively low carbon density, such as grassland and cultivated land, thus leading to a decline in carbon storage. Therefore, changing the land category is an essential way to maintain the carbon balance of the terrestrial ecosystems and increase carbon sinks. The contradiction between the growth of built-up areas, the expansion of cultivated land, and the protection of ecological land in Jilin Province has become increasingly significant in recent years. The interconversion of each category of land is usually accompanied by the carbon emission process, which is not conducive to the formation of carbon sinks. Consequently, the synergy between ecological protection and economic development has become the current development demand of Jilin Province, and the attention should be paid to the protection of grassland and forest land in the future land use planning of Jilin Province. From different climate change scenarios, the SSP1-RCP2.6 scenario can achieve the carbon sequestration goal of Jilin Province the best. This development path has the lowest GHG emissions, and Jilin Province is likely to achieve carbon peaking by 2030. This work quantifies the spatial and temporal evolution of carbon storage and explores the drivers of carbon storage, which are of great theoretical and practical significance to achieving the goal of carbon neutrality. It also provides a reference for exploring the future development path of China or Jilin Province.

### 4.2. Conclusions

In this work, land use in 2030 and 2040 are simulated in different scenarios coupling the PLUS model and InVEST model. The changes in carbon storage under different scenarios from 2000 to 2040 are estimated to explore the spatial-temporal evolution characteristics of CS in terrestrial ecosystems in Jilin Province. The conclusions are as follows.

(1) Cultivated land and built-up areas in Jilin Province increased significantly from 2000 to 2020. The area of built-up areas increased by 7.7%, and the area of cultivated land increased by 0.5%. While the area of forest land, grassland, wetland, and other lands relatively decreased, the area of forest land and wetland decreased by 0.3%, other lands decreased by 0.8%, and the area of grassland decreased by 0.7%. The area of cultivated land and built-up areas will continue to increase in 2030 and 2040, and the area of other land categories will keep decreasing in the SSP2-RCP4.5 scenario. In the SSP1-RCP2.6 scenario, the area of grassland and cultivated land, etc., decreases and is transferred to built-up areas and forest land. The area of forest land, grassland, and other ecological lands under the SSP5-RCP8.5 scenario will decrease, and the area of built-up areas, cultivated land, and other lands used which are used for production and living will increase.

(2) The terrestrial carbon storage in Jilin Province lost 31.8 Tg from 2000 to 2020. Ecological land, such as grassland, forest land, and wetland, is the land category with more serious carbon storage loss. The areas with large losses in CS are the western and southeastern parts of Jilin Province. The carbon storage of the SSP2-RCP4.5 scenario in 2030 and 2040 will be relatively low compared to 2020, but will be slightly higher in 2040 than in 2030. There will be relatively significant changes in carbon storage in the western part of Jilin Province. The SSP1-RCP2.6 scenario shows an overall increasing trend in carbon storage from 2020 to 2040 because of the protection of ecological land. The carbon storage under the SSP5-RCP8.5 scenario gradually decreases, and the reduction in carbon storage in 2030 is mainly concentrated in the western and central part of Jilin Province. The whole region has different degrees of carbon storage decline for 2040, while the area of CS increase is mainly concentrated in the northwestern part of Jilin Province.

(3) The carbon storage in Jilin Province shows a tendency to “increase then decrease” as the altitude rises, with the highest value appearing in the area where the elevation is about 850 m. With the increase in slope, the change in CS is relatively large, and the highest value appears in the area with a relatively high slope. From the perspective of slope direction, the carbon storage is relatively higher on the shaded slopes and lower on the semi-positive slopes. The change in carbon storage from 2000 to 2040 shows a significant consistency with the shift in land use categories.

## Figures and Tables

**Figure 1 ijerph-20-03691-f001:**
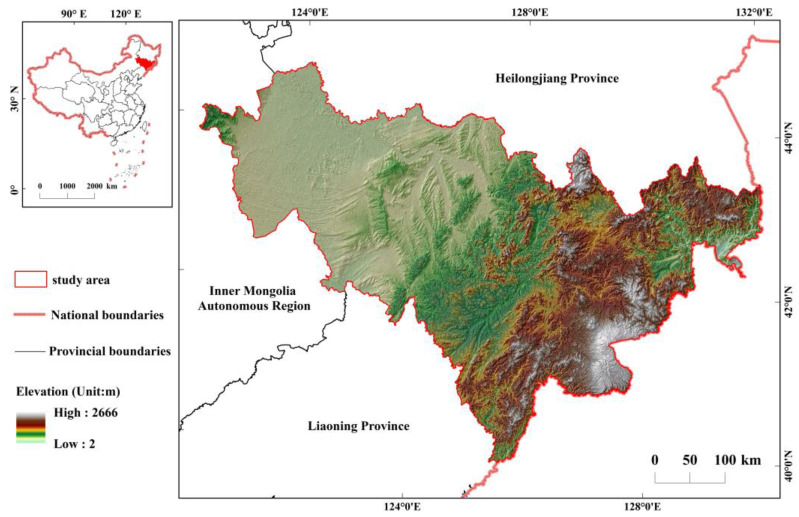
Geographical location and topographic features of Jilin Province.

**Figure 2 ijerph-20-03691-f002:**
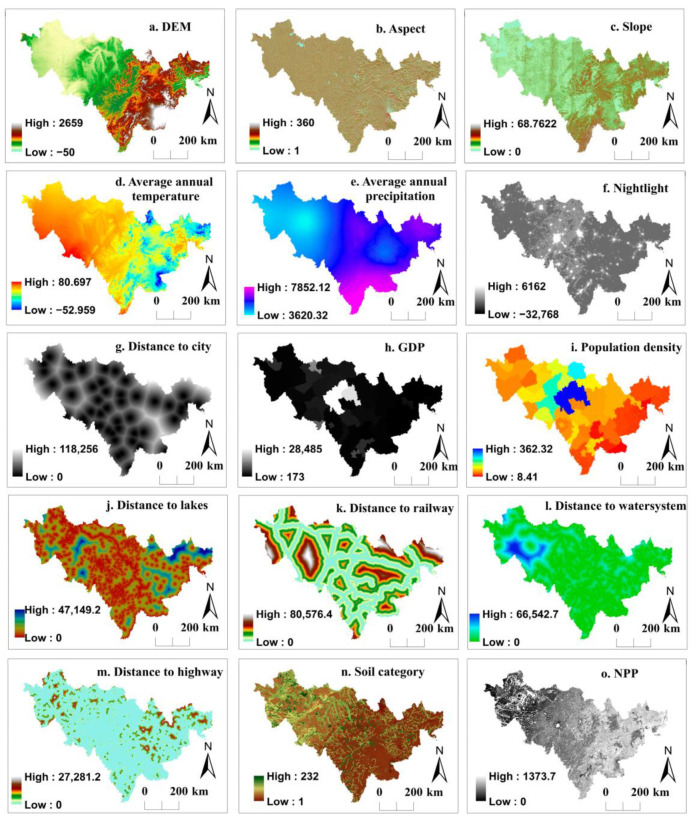
Driving factors.

**Figure 3 ijerph-20-03691-f003:**
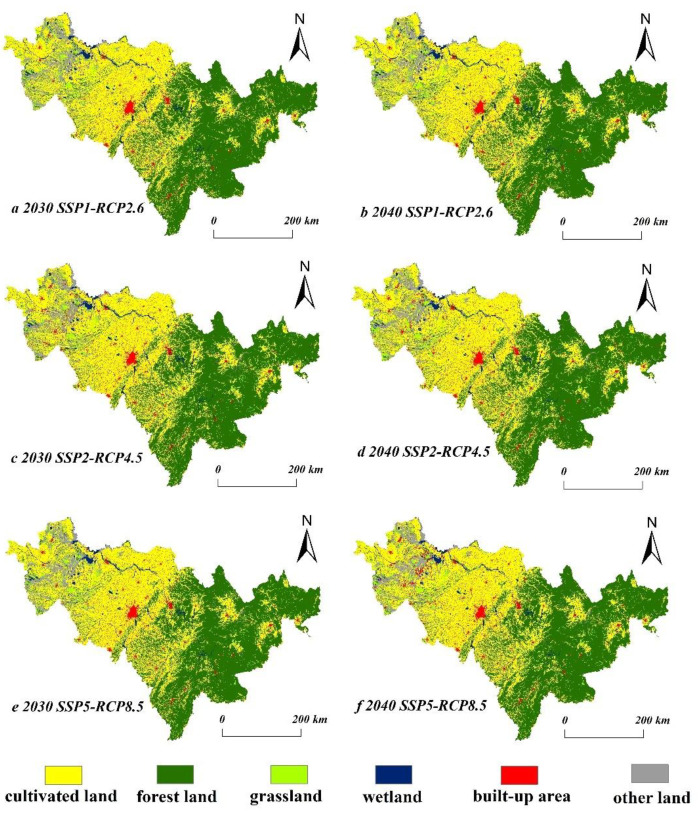
Spatial pattern of future land use under multiple SSP-RCP scenarios in Jilin Province.

**Figure 4 ijerph-20-03691-f004:**
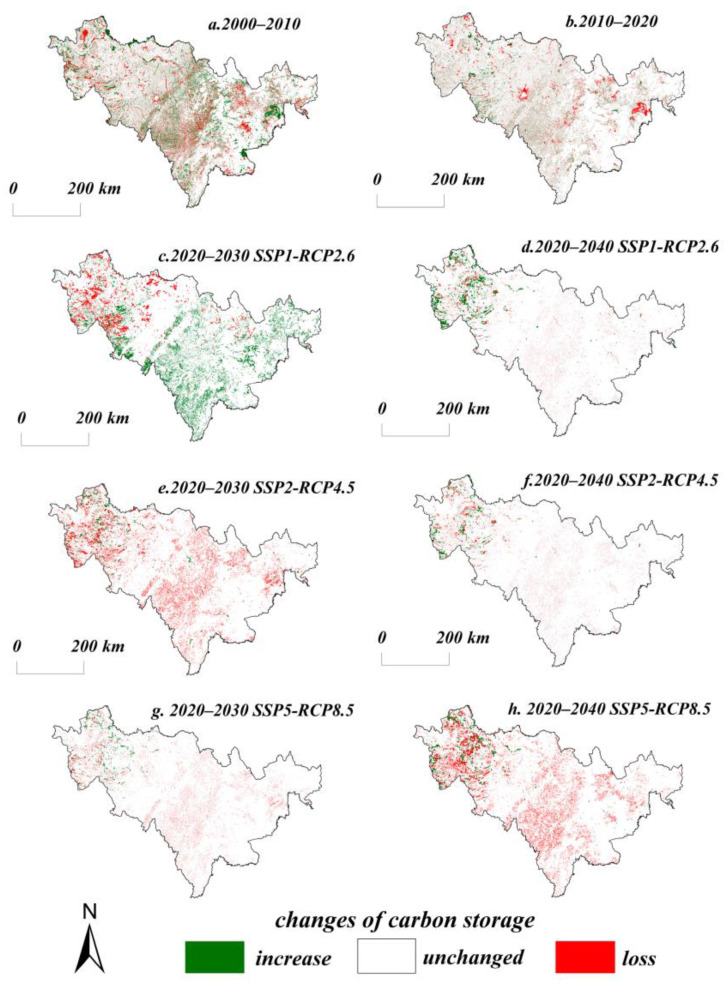
Spatial change in carbon storage in Jilin Province in different periods.

**Figure 5 ijerph-20-03691-f005:**
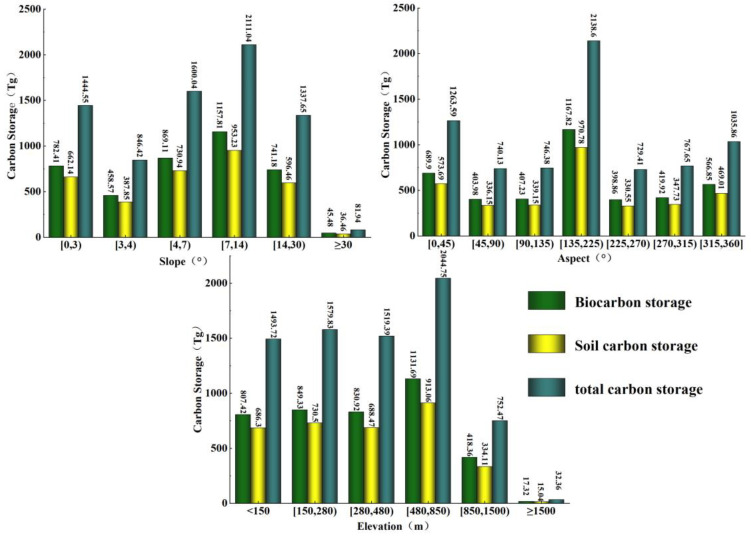
Effects of different topographic factors on carbon storage.

**Figure 6 ijerph-20-03691-f006:**
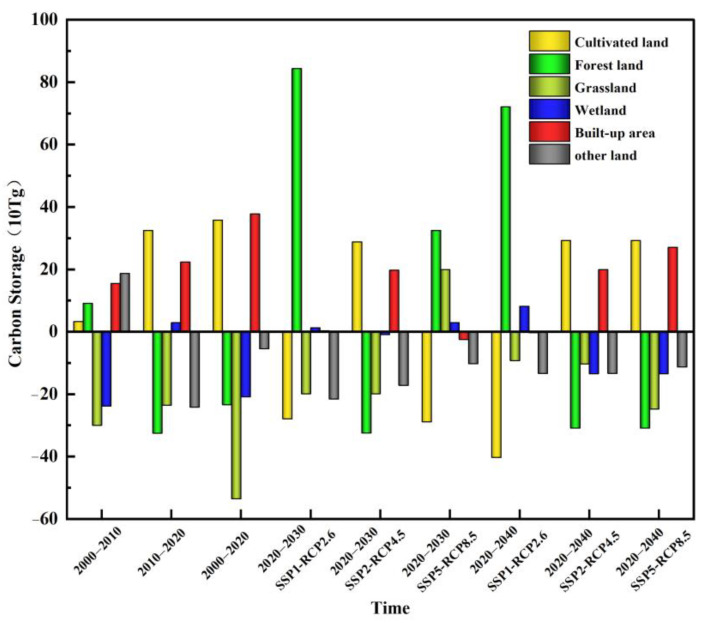
Changes in carbon storage caused by different land use changes.

**Table 1 ijerph-20-03691-t001:** Data content and source.

Data Type	Content of Data	Data Sources
Data of Land Use	Land use data of Jilin Province in 2000, 2010, and 2020 LUH2 dataset	United States Geological Survey (USGS) (http://earthexplorer.usgs.gov/) accessed on 6 May 2021 Land-Use Harmonization (https://luh.umd.edu/) accessed on 11 May 2021
Data of Driving Factor	Average annual temperature, average annual precipitation	National Meteorological Science Data Center (http://data.cma.cn/) accessed on 6 May 2021
	Population density, GDP future climate of SSP-RCP Climate, economic, and population data in RCP scenarios	Geographic Information Monitoring Cloud Platform (http://www.dsac.cn/) accessed on 6 May 2021 NEX-GDDP-CMIP6 database (https://portal.nccs.nasa.gov/datashare/nexgddp_cmip6/) accessed on 20 May 2021 RCP database v2.0 (http://www.iiasa.ac.at/webapps/tnt/RcpDb/) accessed on 21 May 2021
	Soil type, night light data	Resource Environmental Science and Data Center(http://www. resdc.cn) accessed on 6 May 2021
	DEM (slope, aspect)	Geospatial Data Cloud (http: //www. gscloud.cn/) accessed on 6 May 2021
	Basic geographic information data (roads, railways, lakes, water networks, etc.)	Resource Environmental Science and Data Center(http://www. resdc.cn) accessed on 6 May 2021
Carbon Density	Above-ground carbon density, belowground carbon density, soil carbon density	Obtained from the literature and datasets

**Table 2 ijerph-20-03691-t002:** Neighborhood weight parameter settings.

Land Category	Cultivated Land	Forest Land	Grassland	Wetland	Built-Up Areas	Other Lands
2020	0.70	0.30	0.50	0.50	0.80	1.00
SSP1-RCP2.6	0.71	0.40	0.55	0.55	0.90	1.00
SSP2-RCP4.5	0.72	0.30	0.50	0.50	0.90	1.00
SSP5-RCP8.5	0.75	0.25	0.45	0.45	1.00	1.00

**Table 3 ijerph-20-03691-t003:** Carbon intensity values of different land use categories in Jilin Province (Mg/hm^2^).

Land Use Category	Above-Ground Carbon Density	Below-Ground Carbon Density	Soil Carbon Density
Cultivated land	34.0	142.8	150.0
Forest land	59.3	196.9	203.0
Grassland	20.4	94.0	321.6
Wetland	33.1	146.4	157.0
Built-up areas	23.3	120.6	114.3
Other lands	20.8	104.2	250.4

**Table 4 ijerph-20-03691-t004:** Land use transfer matrix of Jilin Province from 2000 to 2020 (km^2^).

Land Use Category (2000\2020)	Cultivated Land	Forest Land	Grassland	Wetland	Built-Up Areas	Other Lands
Cultivated land	65,197.48	4858.72	1032.72	640.88	2948.68	598.04
Forest land	5811.84	76,740.92	1014.68	293.76	223.16	379.28
Grassland	1726.52	1492.60	3567.60	57.40	81.24	938.84
Wetland	670.48	252.36	110.16	2942.44	44.00	839.44
Built-up areas	1716.44	145.00	40.20	25.80	4627.36	32.12
Other lands	1235.44	443.40	869.12	261.76	125.76	8526.96

**Table 5 ijerph-20-03691-t005:** Land use area in Jilin Province under historical period and future multiple scenarios (km^2^).

Time		Cultivated Land	Forest Land	Grassland	Wetland	Built-Up Areas	Other Lands
2020		76,371.40	83,953.96	6636.16	4238.12	8050.84	11,316.84
2030	SSP1-RCP2.6	75,517.08	85,790.96	6179.96	4275.00	8062.32	10,742.00
	SSP2-RCP4.5	75,138.60	85,524.32	6423.60	4480.44	8039.08	10,961.28
	SSP3-RCP8.5	77,253.92	83,247.12	6179.96	4210.00	8816.92	10,859.40
2040	SSP1-RCP2.6	77,266.08	83,280.72	6398.64	3837.12	8823.48	10,961.28
	SSP2-RCP4.5	77,253.92	83,247.12	6179.96	4151.00	8145.32	11,590.00
	SSP3-RCP8.5	77,266.08	83,280.72	6068.36	3837.12	9099.04	11,016.00

**Table 6 ijerph-20-03691-t006:** Carbon storage changes in Jilin Province from 2020–2040 (Unit:Pg).

Time		Biocarbon Storage	Soil Carbon Storage	Total CS
2000		3.910	3.535	7.445
2010		3.918	3.533	7.451
2020		3.910	3.505	7.415
2030	SSP1-RCP2.6	3.931	3.501	7.432
	SSP2-RCP4.5	3.907	3.486	7.393
	SSP5-RCP8.5	3.906	3.496	7.402
2040	SSP1-RCP2.6	3.926	3.506	7.432
	SSP2-RCP4.5	3.906	3.490	7.396
	SSP5-RCP8.5	3.906	3.485	7.391

## Data Availability

Not applicable.

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
