# Peer review of "Assessment of Carbon Storage under Different SSP-RCP Scenarios in Terrestrial Ecosystems of Jilin Province, China"

_ijerph, 2023, doi:10.3390/ijerph20043691_

Round 1

Reviewer 1 Report

Terrestrial ecosystems have absorbed roughly 30% of anthropogenic CO2 emissions over the past decades. Changes in terrestrial ecosystems—for instance, as a result of climate or land use changes—can contribute to changes in carbon storage, which in turn can affect the climate system through the release of greenhouse gases such as carbon dioxide. As such, this study coupled the InVEST model with the PLUS model under different future scenarios to evaluate the evolution characterization of terrestrial carbon storage in Jilin Province from 2000 to 2040 and explored the impact of related factors on it. The results showed that the SSP2-RCP4.5 scenario shows a minimum value of carbon storage in 2030 and a small increase in 2040; the SSP1-RCP2.6 scenario shows an increasing trend of carbon storage from 2020 to 2040; the area of construction land and cultivated land increases and the loss of carbon storage are more serious under the SSP5-RCP8.5 scenario.

Generally, some revision suggestions are listed below:

1) Introduce the importance of Jilin Province in terrestrial ecosystems since the eastern region of Jilin Province is not the whole Jilin Province.

2) Show in details the verification of PLUS model otherwise the results of your research is in doubt. 

3) Discuss your research contribution to greenhouse gas emissions reduction from the perspective of climate change.

Author Response

Dear Reviewer:

Thank you very much for providing so many valuable comments. Here are our responses to your comments:

  • Introduce the importance of Jilin Province in terrestrial ecosystems since the eastern region of Jilin Province is not the whole Jilin Province.

The author’s answer: Thanks for your advice. We have carefully considered the suggestion of Reviewer and make some changes. Jilin Province has an important position in the terrestrial ecosystem of China. We have added a relevant discussion on western Jilin Province in lines 86-88 to complete and introduce the importance of Jilin Province in 1 Introduction.

  • Show in details the verification of PLUS model otherwise the results of your research is in doubt.

The author’s answer: We are grateful for your valuable comments. We have added content about result validation of PLUS model at lines 231 and 236.

  • Discuss your research contribution to greenhouse gas emissions reduction from the perspective of climate change.

The author’s answer: Thank you very much for your comments. We have added relevant content about the contribution of this work to the reduction of greenhouse gas emissions in the context of climate change in 4.1. Discussion in the paper. The details are as follows:From different climate change scenarios, the SSP1-RCP2.6 scenario can achieve the carbon sequestration goal of Jilin Province better. This development path has the lowest GHG emissions, and Jilin Province is likely to achieve carbon peaking by 2030. This work quantifies the spatial and temporal evolution of carbon storage and explores the drivers of car-bon storage, which is of great theoretical and practical significance to achieving the goal of carbon neutrality. It also provides a reference for exploring the future development path of China or Jilin Province. ( Line 488-494 in the paper)

Reviewer 2 Report

In this paper, the PLUS model with the InVEST model was used to investigate the spatial-temporal partitioning and evolutionary characteristics of carbon storage in Jilin Province. The subject of the manuscript concerns a very important topic which is global warming and takes into account different SSP-RCP scenarios to estimate and simulate terrestrial ecosystem carbon storage in China's province. The work is suitable to be published in the International Journal of Environmental Research and Public Health. However, the work is poorly written. The manuscript needs improvement in the English language, numerous typos, missing spaces, and font changes are visible. Formulas and tables are written carelessly. The quality of Figs is insufficient. 

Author Response

Dear reviewer:

Thank you for your good comments. We have made corrections according to your comments. We corrected some sentences and words in the article and added the missing spaces. We set the text content to a uniform font and adjusted the formulas and tables in this paper. We also made some changes to the text in the images. All changes to this paper have been marked using the “Track Changes”.

Reviewer 3 Report

Please revise manuscript follow the comments as shown in attach file.

Author Response

Dear reviewer:

Thank you for providing such detailed comments on this paper. We have made changes to the corresponding lines in the paper (some lines may have changed due to some additions). All changes to this paper have been marked using the “Track Changes”.

Here are some changes we made to this paper.

  1. We have added the missing spaces with reference to your comments.
  2. For comments 2, 3, 4, 7, and 12, we have added the full names of the abbreviations.
  3. We changed "this paper" to "this work" in the paper.
  4. We have removed unnecessary conjunctions and adjusted some sentences.
  5. For comments 4, 10, and 15, we added details about Figure 2, Table 2, and Table 3.

Round 2

Reviewer 1 Report

Accept